# Examining the Impact of Virtual Animal Stimuli on College Students’ Affect and Perception of their Academic Advising Experience

**DOI:** 10.3390/ani13091522

**Published:** 2023-05-01

**Authors:** Elizabeth A. Johnson, Sheetal Survase, Peter B. Gray

**Affiliations:** 1Department of Anthropology, University of Nevada, Las Vegas, NV 89154, USA; peter.gray@unlv.edu; 2School of Public Policy and Leadership, University of Nevada, Las Vegas, NV 89154, USA; survase@unlv.nevada.edu

**Keywords:** anthrozoology, human–animal interaction, well-being, virtual, companion animal, intervention, mental health benefits, human–animal bond, student connectedness, social support

## Abstract

**Simple Summary:**

Student’s mental health and well-being is a growing concern on college campuses, yet barriers such as stigma, lack of awareness, or long waitlists for services can prevent students from seeking help. While the benefits of animal-assisted interventions (AAI) have been established in the literature, accessibility of animals within public spaces or interactions with some populations can present limitations to the use of AAI. This research examines the impact of animal visual stimuli as an alternative to help address the limitations to using AAI. In this study, college students are presented with randomized one-minute videos of either nature, wild animals, companion animals or a control condition prior to virtual advising appointments to measure the impact of the type of stimuli on their well-being, perceptions of their advisor and their university. Results indicated all animal videos increased well-being measures and companion animal stimuli influenced the student’s perception of the advisor but had no impact on student’s perception of the university. The findings support that computer-mediated interventions fill an important service gap to improve well-being outcomes when individuals do not have access to traditional forms of AAI assistance and can ultimately have a broader impact outside of advising and the university.

**Abstract:**

The benefits of animal-assisted interventions (AAI) involving animals in therapy are widely accepted. The presence of animals in therapy can decrease a patient’s reservation about therapy and promote a sense of comfort and rapport during the therapy process. Using survey data from college students (*n* = 152) attending a large public four-year institution, this study is the first to investigate the benefits of virtual animal stimuli during academic advising appointments. It posits that exposure to virtual animal stimuli can influence positive mental health and well-being in academic advising settings. Specifically, the research questions explored how different types of video content influence students’ affect and how virtual animal stimuli impact students’ perception of their advisor and university. College students were randomly assigned to watch one of four types of virtual stimuli (wild animals, companion animals, nature, and a control) prior to their advising session. Subjective measures were collected at baseline and after the advising session. Results indicated animal stimuli increase positive affect, and companion animal stimuli influence the student’s perception of the advisor. This study supports the notion that companion animal videos positively impact students’ well-being and interactions with their advisors and may have broader implications beyond the academic setting.

## 1. Introduction

Findings from The American College Health Association (ACHA) College Health Assessment II: Reference Group Executive Summary Spring 2018 support students perceive stress as an impactor to their individual academic performance [1]. Student mental health has been a growing concern for U.S. colleges and was worsened by the COVID-19 pandemic [2]. Student stress has important implications for student retention and is a concern for U.S. colleges [3]. As a response to the mental health challenges faced by students, colleges have implemented programs aimed at supporting student well-being and promoting attitudes towards their institution [4]. Programs can include mental health counseling and other non-traditional forms of support which have increasingly risen in various modalities. However, attendance rates for campus health services remain low [5].

Animal-assisted intervention (AAI) is one such non-traditional program implemented on campuses [6]. According to the International Association of Human-Animal Interaction Organization (IAHAIO), AAI is a goal-oriented and structured involvement that intentionally incorporates animals in health, education, and human service for the purpose of therapeutic gains in humans [7].

AAI’s therapeutic gains come from incorporating human–animal interaction (HAI) to improve physical, social, emotional or cognitive functioning [8]. Research suggests that the animal in AAI acts as a helping agent in an ameliorative process or to affect the course of people’s lives [9]. A growing body of literature on the benefits of AAI on well-being [10,11,12,13,14] has led to widespread acceptance of animals for therapeutic services among the public. However, there are limitations to the accessibility of live animals in public spaces (e.g., animal-related accidents, financial constraints) or with certain populations (e.g., allergies, infections) [15]. Additionally, COVID-19 has improved information and communication technology that has allowed for an uptick in the shift to online support services post-pandemic [16,17] and a recent study indicates virtual AAI interventions can have beneficial effects on well-being [15].

Accordingly, this study focuses on animal visual media as a complementary tool, like that of AAI, in shaping participants’ well-being in non-clinical interventions in academia (academic advising). Specifically, using an evidence-based causal design, this study investigates the possible health benefits from visualization of various types of animals prior to an established and required model of academic advising.

Advising has been extensively cited as a necessary element of students’ success, i.e., retention, persistence, and degree completion [18,19,20,21]. Thus, connection between students and advisors becomes fundamental for academic advising center goals, missions, and learning outcomes. Interpersonal attachments that humans form are fundamental for human motivation [22]. In academic advising, a student–advisor connection is crucial to guide students’ intrinsic motivation and confidence levels in their education. Students’ perception of support by the advisor has a more substantial impact than general university support, and these interactions can be significant predictors of educational outcomes [23]. Adding a virtual component to advising appointments may be a complementary way to impact affective well-being and social connections through a sense of belonging [24]. Hence, this study presents a non-traditional opportunity to implement a program to increase positive affect and promote well-being within a required academic advising appointment. Using an anthropological framework, the aim of this study was to understand preemptive care within the academic setting, to assess the value of animal content for emotion regulation and perceived participant impact, and to create a non-traditional model of care for various institutional settings that mitigates a growing health concern.

### Visual Stimuli and Emotion Regulation

Human activities have impacted much of the earth’s terrestrial surface, and in doing so, contact with animals is inevitable [25]. This idea aligns with the Biophilia hypothesis, which posits that humans have an adaptive response to the natural world (nature and animals). Further, cats and dogs, domesticated for thousands of years, play various social roles from companionship and emotional support to social lubricants. In this way, pets as a subject in visual stimuli may provide different patterns of emotion and social support (social bonding and a sense of belongingness). The variable nature and animal stimuli may then be emotionally salient to students and can upregulate a positive response essential to health and well-being.

Using video content incorporates the idea of extrinsic emotion regulation (ER), in which a provider’s deliberate goal is to change or influence a recipient’s emotional trajectory [26,27,28]. ER is used to relax, cope with stress, and manage or avoid negative feelings [29]. Additionally, digital forms of ER in various modes provide prosocial behavior, i.e., self-reflection and social sharing [30]. In considering the practitioners’ role, access to an emotional support animal can provide emotional benefits for a diagnosed mental health condition with a goal for each animal to improve the individual’s well-being. However, not everyone has access to in-person AAI, and not all programs have the physical means to provide animals in-person; therefore, computer-mediated interventions may fill an important service gap in the context of improving mental health outcomes when individuals do not have access to physical AAI. In this way, this intervention can be considered similar to a virtual AAI intended to increase positive affect and emotion and ultimately to improve well-being.

Using diverse stimuli (nature, wild animals, companion animals, and control: random dot pixel pattern of static) in a pre- and post-experimental design can establish causal inferences about the effect of the type of stimuli on measures of well-being in humans. A growing body of literature indicates that greater exposure to natural environments is associated with better health and well-being outcomes [31], including affect and mood [32]. Indirect exposure has been documented to positively affect participants’ physiological health [33,34]. For example, Laumann, Garling and Stormark found that videos of nature reduced heart rate in comparison to urban environments [33]. Additionally, cardiac interbeat intervals (IBI) counts suggest videos depicting natural environments had a relaxing effect on autonomic functions and elicited reduced physiological arousal [33]. Studies suggest that virtual stimuli remain limited in their ability to enhance physiological responses compared with actual nature; however, virtual nature is an effective alternative to improve psychological benefits for those who cannot easily access real nature [35].

Several studies support the association between BBC Planet Earth clips featuring wild animals and increased positive emotions and decreased negative emotions [36,37,38]. Humans have an innate ability to relate to and empathize with animals. Attributing human-like thoughts, feelings, motivations, and beliefs towards animals is suggested to have evolved from successful hunting and gathering strategies by our ancestors [39,40]. Scholars suggest our desire for pets and kinship relations makes this response stronger in our interactions with domesticated animals [40,41], this phenomenon is more prominent in urban populations [14]. Further, companion animals have infantile-like features, commonly referred to as baby schema which have been found to activate the nucleus accumbens, a key structure of the mesocorticolimbic system that is linked to the anticipation of reward [42]. This bond may upregulate when an individual sees an image of a companion animal even if it is not their own pet.

Research supports that mediated exposure to animal content may result in similar outcomes found in AAI [43]. Thelwell found that among participants who watched videos of dogs and interacted with dogs both reported a reduction in negative affect and increase in positive affect [44]. Kogan et al. found that veterinary students who took short breaks during their pharmacology class to watch pet videos (including puppies and kittens) had positively affected mood, interest in the material, and self-reported understanding [45].

Rossbach and Wilson [46] found that respondents perceived people in pictures with animals to be more relaxed and happy than people in pictures with flowers. Wells and Perrine found that students perceived their faculty to be friendlier, positive, less threatening, and their office to be more comfortable when shown photographs of their office with either a dog or cat near their desk [47]. These findings are supported by previous research [48].

Thus, we can expect companion animal visual stimuli prior to engaging with an advisor to provide positive emotions and act as a social lubricant. By introducing a one-minute video as extrinsic digital ER, advisors may deliberately influence participants’ affect. The content allows the individual to reappraise their emotions going into a meeting, i.e., self-soothe negative thoughts and control attention in a rewarding manner, and thus prime a positive social environment for the participant. The content is followed by their advising appointment, and it is this interaction with the advisor that influences the student’s perception of the university.

We sought to investigate changes in students’ affect based on the type of video content watched prior to an academic advising session as well as changes in students’ perception of their advisor and university. Ultimately, we tested the following hypotheses:

**H_1_:** Exposure to videos of nature, wild animals, and companion animals will increase positive affect in comparison to control conditions.

**H_2_:** Exposure to videos of nature, wild animals, and companion animals will decrease negative affect in comparison to control conditions.

**H_3_:** Companion animal stimuli prior to an advising session will increase students’ positive perspectives of their advisor in comparison to the control and nature.

**H_4_:** Wild animal and companion animal stimuli prior to an advising session will not increase students’ positive perspectives of the university after advising sessions in comparison to the control and nature.

## 2. Materials and Methods

This study was approved by the University of Nevada, Las Vegas Social/Behavioral Internal Review Board (UNLV-2022-268 and 24 August 2022). Informed consent was obtained from all subjects involved in the study.

### 2.1. Advisor Recruitment and Participant Screening

Advisors at the Academic Success Center were recruited to participate in this study by word of mouth between 24 August 2022 and 20 January 2023. All advisors were directed to complete an initial screening survey using Qualtrics (www.qualtrics.com, accessed between 24 August and 1 November 2022). Eligibility to participate included being an Academic Success Center advisor and having their camera on during the entirety of the appointment. All advisors were informed that this study provides a USD 20 Amazon gift card in compensation and their participation is voluntary. Advisors who chose to participate attended a training session with the researcher and participated in a mock virtual advising session. Advisors were informed of required tasks and provided the following materials to students: pre-email invitation to student participants, text providing students with each Qualtrics survey link on the Webex platform, coding system to track students, information regarding confidentiality, debriefing statement in appointment follow-up email, and an optional Qualtrics survey to note their perception of student behavior. At the end of training, advisors verbally confirmed that they felt comfortable performing the tasks of the study and were encouraged to ask questions about this study at any time during the research.

Advisors were informed that the intention of this study is to evaluate how watching video content at the beginning of a virtual academic advising session may affect social interaction and influence affect. The advisors were also informed that the Qualtrics survey at the beginning of the meeting should take no longer than 10 min to complete and that during this survey, students who participated will watch a short video as part of the study procedures. All advisors were blind to the video material during each session. The second survey at the end of the meeting did not have video content and should not have taken longer than 5 min to complete.

### 2.2. Student Recruitment and Participant Screening

Student participants were recruited in Las Vegas, NV, USA via participating UNLV Academic Success Center (ASC) advisors between 26 August 2022 and 20 January 2023. Recruitment materials described the study as an academic project to evaluate students’ emotion, mood and perception of advisor and their meeting after watching video content. The study also advertised an incentive: participants were able to opt-in to a raffle for a USD 100.00 Amazon gift card after they had completed the study. Advisors were instructed to email students prior to an advising session to inform the student of inclusion criteria as well as to request a response to participate in the study. Participants were informed that any decision to participate in this study would not affect their relationship with their advisor or the advising center. Interested student participants were required to meet the inclusion criteria which encompassed being present at their virtual Webex appointment.

A total of 455 virtual appointments were conducted during the time period of the study. Based on G*Power 3.1 software [49], the a priori sample size was determined by using an ANOVA: Repeated measures statistical test of four conditions and two measures, effect size (Cohen’s f) of 0.25 (medium), probability of alpha error of 0.05, and power of 0.95. The minimum sample size required was 76. Therefore, approximately 19 participants would be needed in each condition. Advisors were able to recruit 156 student participants. During the timeline of this study the majority of students were first-time freshman, sophomore and Adult Learners (individuals who are typically 25 years or older and are pursuing some form of post-secondary education), seeking advice on course selection, probation, suspension or major exploration.

### 2.3. Participants

As of fall 2021, the advising center in this study was home to 2119 undergraduate students and 785 non-degree seeking students with a diverse set of backgrounds and educational needs. The center selected for this study requires all students within the major to meet with an academic advisor at least once a year. Of eight advisors employed in the ASC professional academic advising role, five advisors were recruited to participate in the study. Selected advisors served in professional advising roles where their primary role is to advise students. Of the five advisor participants, two were not retained during the months of the study. Subsequently, two new ASC advisors were selected to participate (*n* = 7). The ASC is one of the 12 advising centers at the University of Nevada, Las Vegas, NV, USA. This particular advising center’s advisors meet with students who are considered exploring, non-degree seeking, returning, on university probation or suspension and major pathways students (see, University of Nevada, Las Vegas Academic Success Center Advising, n.d.) [50].

Participating advisors held to the Developmental Advising approach, wherein advisors and advisees entered into a process-oriented relationship, in which the main focus is developing academic plans for the realization and fulfillment of students’ goals as well as developing student’s decision-making and problem-solving skills necessary to enhance their learning experience [51,52]. Developmental advising focuses on the growth of the student and works in tandem with students to raise awareness of the link between education and life and set realistic academic goals. Developmental advising is a fully student-centered approach to student development and has shown to positively increase student retention and success rates [18]. The vision of the center is to employ best academic and developmental practices that positively impact the lives of students at the institution.

### 2.4. Intervention Treatments

Visual content for each of the videos was sourced from BBC Planet Earth [53] and various Youtube.com videos under the search “TV white noise,” “cat,” and “dog” accessed within the public domain in the month of January 2022. The content was compiled into four randomized one-minute videos with uniform instrumental sound across each stimulus. The videos created include control, nature, wild animals, and dog and cat content. The control included a random dot pixel pattern of static. This control was chosen to be a neutral option in comparison with the other visual stimuli. The nature content included nature scenes empty of humans or animals that were located outside the state of the university. The nature content is specifically void of animals and people, as research has suggested that green spaces can aid in well-being [54] and can be specifically beneficial for urban dwellers [55]. Wild animals in nature were incorporated as another stimulus and were selected for their ability to be aesthetically pleasing in U.S. culture [56]. Finally, the dog and cat content included various dogs and cats in nature settings.

### 2.5. Measures

The following survey instruments were administered repeatedly in the pre-post survey format to measure the subjective dimension of affect.

The *Positive and Negative Affect Schedule* (PANAS) [57] is a self-administered questionnaire that consists of two separate scales of 10-items each to measure two dimensions of emotional experience: positive affect (PANAS-P) and negative affect (PANAS-N). Students were asked to rate all 20 adjectives that assessed participants’ positive feelings on the PANAS-P (e.g., proud, inspired) and negative feelings on PANAS-N (e.g., nervous, guilty) that best reflects their experience presently or in the past 40 min. Items are rated on a five-point Likert scale: 1—Very slightly, 2—A little, 3—Moderately, 4—Quite a bit, 5—Extremely.

The *modified Differential* Emotion *Scale* (mDES) [58] consists of 20 items, each describing a specific emotion or feeling. Similar to the PANAS, mDES consists of two subscales of 10 items each to measure positive emotions (mDES-P) (e.g., awe, wonder, amazement) and measure negative emotions (mDES-N) (e.g., disgust, distaste, or revulsion). Students were asked to rate the frequency with which they experienced each emotion on a 5-point scale, ranging from 0 (never) to 4 (most of the time) [59,60]. This scale was modified to measure frequency rather than intensity to address the research question and gain a more complete understanding of an individual’s emotional experience in response to the intervention. Further, frequencies have been found to be more stable over time than intensity measures [61]. Including both frequency and intensity (PANAS) measures can provide a more nuanced picture of a participant’s emotional experience over time [62], which is particularly useful in emotion regulation.

Similarly to the PANAS, students were asked to best reflect on the frequency of their experience of adjectives on the mDES-P and mDES-N presently or in the past 40 min. This temporal frame helped capture immediate effects of the intervention on participants’ affect while minimizing confounding effects of external factors over longer periods. Evidence supports the idea that individuals can change their emotions in a relatively short amount of time using emotion regulation strategies [63,64,65,66].

While both the PANAS and mDES measure affect, they are widely used scales that differ in their scope and focus. Therefore, the scales were not combined into a composite score. The PANAS is used to assess activated forms of positive and negative affect and the mDES is used to assess a broader array of discrete positive and negative emotions. The mDES was specifically created to be more encompassing of positive emotions, while the PANAS exclusively targets high-activation positive affective states [67,68]. Further, the scales have been used together in previous research [69].

The *University Belonging Questionnaire* (UBQ) [24] consists of 24 items that measure the sense of belonging that students experience in a university. Respondents were asked to rate their agreement with each statement on a scale ranging from 1 (strongly disagree) to 4 (strongly agree). To address the study’s aim and prevent response fatigue, the survey focused only on two factors measuring university support and acceptance, and faculty and staff relations. The 12 items on factor measuring university affiliation were excluded. Out of the eight items related to university support and acceptance, six were utilized in the study. The two items excluded related to diversity at the institution to keep the study within its intended scope. Three of the four items were adapted from the factor measuring faculty and staff relations. Statements including words such as ‘faculty/ staff’ were replaced by the word “advisor”, and statements were made shorter and concise to minimize respondent burden. For example, “I believe that a faculty/staff member at my university cares about me” was changed to “I believe my advisor cares about me.” Another example, “I feel connected to a faculty/staff member at my university” was changed to “I feel connected to my advisor.” Additionally, as all the advising sessions relevant to this study were conducted virtually, one item relating to the virtual setting was included in the questionnaire: “I feel comfortable with the virtual advising setting.” This variable was included in the analysis to determine whether virtual advising had any effect on the student’s response to the video stimuli. As such, this questionnaire would be considered an adaptation of the UBQ specific to the purposes of this study. Respondents were asked to rate their agreement with each statement on a scale ranging from 1 (strongly disagree) to 4 (strongly agree).

To measure internal consistency and reliability of the above instruments this study utilized Cronbach’s alpha. The alpha values for the PANAS-P (baseline [α = 0.915], post-intervention [α = 0.920]), PANAS-N (baseline [α = 0.876], post-intervention [α = 0.888]), mDES-P (baseline [α = 0.947], post-intervention [α = 0.941]), mDES-N (baseline [α = 0.895], post-intervention [α = 0.914]), and UBQ (baseline [α = 0.979], post-intervention [α = 0.981]) were above 0.70, a value considered acceptable for research purposes [70].

Item measuring student feedback of the video stimuli provided, participants were asked to report on items such as, “How does this video make you feel?” to which they could answer positively, negatively, or indifferently. Participants then answered “yes/no” to the following questions: “Did you like the content in this video?”, “Would you be interested in watching similar videos before each advising session?”, and “Does this video increase your interest in the upcoming advising session?”. This was included in the study to provide insight on a student’s feelings toward the video assigned. These items were only included after the treatment video.

### 2.6. Procedures

Prior to the intervention, advisors completed an informed consent form and a 30-min training on approaching the interaction. The advisor was asked not to explicitly ask their students about the video content during the intervention. The intervention incorporates a pre-email, a WebEx appointment, two surveys (pre-post), a questionnaire, a set of pre-survey questions and a debriefing email. Each advisor was asked to send pre-email invites to participants in the study who plan to attend a WebEx meeting (virtual platform). After the participant agreed to participate via the pre-email message, the advisor opened WebEx with their camera off and posted a welcome message, uniform instructions with a participant code for anonymity between surveys, and a survey link in the chat section. Preventing the student from seeing the advisor before the interaction was an attempt to prevent a visual emotional response prior to the completion of the survey.

Student participants were directed to complete the initial screening survey using Qualtrics. Once students completed the informed consent, the survey directed them to complete additional questions about their sociodemographics (e.g., marital status, education level, sex, age), questions to measure student affect and belongingness, and a video and questions related to interest towards the video material. The pre-survey consisted of two scales, the Positive and Negative Affect Schedule (PANAS) and the modified Differential Emotion Scale (mDES). An adaptation to the University Belonging Questionnaire (UBQ) was also included prior to a randomly selected video stimulus and questions related to the stimulus. Once the student completed the pre-survey and watched the video, they notified their advisors. At this point the advisor turned on their camera and proceeded with the appointment. At the conclusion of the appointment, advisors provided the secondary survey in the chat. The post-survey consisted of the two scales, the PANAS-SF and the mDES, and the adaptation of the UBQ in the opposite order of the pre-survey. The estimated total time to complete the pre- and post-surveys was about 15 min. After completion of each advising session, the advisor could opt in to complete an advisor perception survey if the student mentioned video content or if the advisor had comments relevant to the study during their advising session. If the student mentioned the video content, the advisor was asked to describe how the student brought up the video content in the advising session and to note if the comment was positive, negative, and indifferent. Additionally, another open-ended question asked advisors to provide any other comments that may be useful to the study.

### 2.7. Statistical Analysis

The Statistical Package for the Social Sciences (SPSS, version 28.0, IBM software) was used for statistical analysis. Scores for items on the subscales of the PANAS and mDES, mDES-P and mDES-N were summed and the mean scores for individual responses were obtained, forming four new scaled variables measuring students’ positive affect (PANAS-P), negative affect (PANAS-N), frequency of discrete positive emotions (mDES-P) and discrete negative emotions (mDES-N).

Repeated measures ANOVA was selected as the overall model (all visual stimuli) for response across time to understand each comparison and the effects of the different video treatments on participants’ response on affect and emotion. As there is no current gold standard for test of normality, and Log10 or square root transformation methods did not significantly change the data distribution, the following data were determined to be normally distributed for repeated measures ANOVA if it fell between the absolute skewness and kurtosis values of −2.0 and 2.0. The values for asymmetry and kurtosis between −2 and +2 are considered acceptable in order to prove normal univariate distribution [71]. Survey mean values of the pre- and post-survey were used in a repeated measures ANOVA as within-subject variables to compare treatments as between-subject factors and to consider whether potential covariates might yield different pre–post responses between participants. Following this, a correlation analysis was run to understand the relationships between each scale.

Due to the adaptations made to the UBQ for this study, a Principal Components Analysis was used to determine if a distinction could still be made between the two factors of interest—university support and acceptance, and faculty and staff relations. A student’s belongingness was calculated to a sum score of the UBQ scale for both pre- and post-survey. Kruskal–Wallis was employed to test if there was a difference between treatments. Post hoc Mann–Whitney U tests with a Bonferroni-adjusted alpha were utilized to answer hypothesis H_3_ and H_4_. To reduce the chances of false-positive results, Bonferroni correction was used when multiple pairwise tests were performed. For a deeper analysis, all items were tested individually using the same methodology.

χ-squared tests were used to compare pre-survey participants’ responses to items after watching the randomized treatment. Items related to student perceptions of virtual advising were not included in the post survey. We present the findings of the analysis in Results below.

## 3. Results

### 3.1. Participants

A total of 156 responses were collected via Qualtrics. Often individuals responding to surveys experience survey fatigue. As our pre–post survey consisted of over 50 questions and provided an opportunity to win a USD 100 Amazon gift card, there was a possibility that participants would not answer questions thoughtfully. As suspected, there were four participants who provided false data points (e.g., no variation in survey data, i.e., all “3” responses) or inconsistencies between responses to similar questions. As such, we removed these individuals to improve the quality of the survey data (*n* = 152). The analysis consists of four videos: nature (*n* = 38), wild animal (*n* = 38), companion animals (*n* = 38), and a control (*n* = 38). The descriptive information for our final sample (*n* = 152) can be found in Table 1. Univariate analyses did not find any differences in participants’ responses based on their socio-demographic characteristics including age, gender, ethnicity/race, class level, and major.

In this study, seven advisors received extensive training in implementing the study’s requirement within their general advising practice. As such, for the purposes of the study it can be expected that advisors conducted themselves the same across students and were not considered a covariate.

The correlational analysis supported some strong correlations between PANAS and mDES measures. This was expected considering the similar nature of the constructs measured by the two scales, see Table 2.

### 3.2. PANAS-P and mDES-P

To address the first hypothesis, a repeated measure ANOVA was conducted to determine whether there is an effect of the treatment on positive affect (PANAS-P). Mean, standard deviation and range can be found in Table 3. The results indicate that there is a significant difference between treatments, (F (1, 148) = 14.664), *p* < 0.001 n2p = 0.090, observed power = 0.967. This finding rejects the null hypothesis and concludes there is a difference between treatment and positive affect measures (Interested, Excited, Strong, Enthusiastic, Proud, Alert, Inspired, Determined, Attentive and Active). The effect size was medium [72] and the observed power was 0.967, indicating a 96.7% chance of detecting a significant effect given the sample size, effect size, and alpha level. A further post hoc comparison using the Tukey HSD test indicated that the mean score for the companion animal treatment was significantly different than the control treatment (*p* = 0.038). However, the nature and wild animal treatments did not significantly differ from the companion animal treatment (*p* = 0.969, *p* = 0.959, respectively) or the control treatment (*p* = 0.113, *p* = 0.126, respectively).

Following the PANAS-P, the mDES was evaluated using a repeated measures ANOVA to determine whether there is an effect of the treatment on positive emotions (mDES-P). Mean, standard deviation and range can be found in Table 4. The results indicate there is a significant difference between treatments, (F (1, 148) = 32.690), *p* < 0.001 n2p = 0.181, observed power = 1.0. The effect size was large [72] and the observed power was 1.0, indicating that there was a 100% chance that the results would have come out significant. This finding rejects the null hypothesis and concludes there is a difference between treatment and positive affect measures (amused, fun-loving, or silly; awe, wonder, or amazement; grateful, appreciative, or thankful; hopeful, optimistic, or encouraged; inspired, uplifted, or elevated; interested, alert, or curious; joyful, glad, or happy; love, closeness, or trust; proud, confident, or self-assured; serene, content, or peaceful). A further post hoc comparison using the Tukey HSD test indicated that the mean score for the wild animal and companion animal treatments were significantly different than the control treatment (*p* = 0.012, *p* = 0.002, respectively). However, the nature treatment was not significantly different from the control treatment (*p* = 0.174).

Exposure to videos of nature, wild animals, and companion animals will increase positive affect in comparison to control conditions. In the case of the PANAS, we would partially accept H_1_ that the companion animal treatment increased students’ positive affect compared to the control; however, nature or wild animals did not differ from the control. Similarly, to the PANAS, mDES findings would partially support H_1_. In this case, wild animal and companion animal treatments increased students’ positive emotion compared to the control; however, nature did not differ in comparison to the control.

### 3.3. PANAS-N and mDES-N

Following the analysis of the positive affect and emotion, a repeated measure ANOVA was conducted to determine the effect of the treatment on negative affect (PANAS-N) which provides information for H_2_. Mean, standard deviation and range can be found in Table 4. The results indicate significant differences, (F (1, 148) = 62.562), *p* < 0.001, n2p = 0.297, observed power = 1.0. The effect size was large, and the observed power was 1.0, indicating that there was a 100% chance that the results would have come out significant. Subsequent analysis of post hoc comparisons using Tukey HSD indicate there is no difference between nature, wild animal and companion animal treatments in comparison to the control (*p* = 0.958, *p* = 1.0, *p*= 0.979, respectively).

Similarly, the repeated measure ANOVA to determine the effect of the treatment on negative emotions (mDES-N) indicated there was a significant difference, (F (1, 148) = 74.168) *p* < 0.001, n2p = 0.334, observed power = 1. The effect size was large, and the observed power was 1.0, indicating that there was a 100% chance that the results would have come out significant. However, subsequent analysis of post hoc comparisons using Tukey HSD indicate there is no difference between nature, wild animal and companion animal treatments in comparison to the control (*p* = 0.996 *p* = 0.878, *p*= 0.996, respectively). Mean, standard deviation and range can be found in Table 3.

H_2_ Exposure to videos of nature, wild animals, and companion animals will decrease negative affect in comparison to control conditions. Based on our findings, hypothesis 2 was rejected.

### 3.4. UBQ

A Principal Component Analysis of the 10 UBQ items for each section was completed (pre-survey and post-survey, respectively). For both sections, all statements loaded strongly into one factor which explained 84.35–85.37% of variance. PCA was attempted using Varimax rotation and the phenomenon remained, all statements loaded above 0.800. Additionally, there was no difference when testing each of the treatments. While only one factor was extracted, this study maintained university support and advisor relations as distinct factors. The definition of sense of belonging for the purposes of this study encompasses a range of experiences including students’ perceived social support on campus, a feeling of connectedness, the experience of feeling cared about, accepted, valued by their campus. Within the context of this study, it is arguably important to separate the two factors of university support and advisor relations. Based on the literature [73,74,75] and authors’ professional advising experience, the nature of the advisor–student relationship warrants a separate evaluation from university support. Students may feel more connected with their advisor and less connected with their university, especially when approaching their advisor with concerns or challenges they are experiencing at the institution. Therefore, it is crucial to evaluate university support and advisor relations as separate factors in order to understand the impact of the video stimuli on each variable and to gain insight into the effectiveness of the intervention.

A Kruskal–Wallis test showed that all treatments were not significantly different between the pre-survey group but was significantly different between the post-survey groups, (H(3) = 3.761, *p* = 0.288, H(3) = 10.111, *p* = 0.018, respectively). H_3_ does not include wild animals, so another test was performed only using three treatment types (*n* = 114). Results were similar H(2) = 3.143, *p* = 0.208, H(2) = 8.106, *p* = 0.017, respectively). Post hoc Mann–Whitney U tests using the Bonferroni-adjusted alpha level of 0.25 (0.05/2) were used to compare pairs of groups. When comparing the companion animal treatment group (Md = 40) to the nature treatment group (Md = 34), there was a significant difference between the post survey group (U = 492.500, z = −2.453, *p* = 0.014). When comparing the companion animal treatment group to the control treatment group (Md = 34), there was a significant difference between the post-survey group (U = 488.000, z = −2.505, *p* = 0.012).

Given that the H_3_ hypothesis was related to students’ advisors, it was important to explore this concept further. There are three advisor-specific questions:I feel that my advisor has appreciated me.I feel connected to my advisor.I believe that my advisor cares about me.

For all three questions we employed a Kruskal–Wallis test with only the three treatments listed in the hypothesis (*n* = 114). All pre-survey treatment groups had no significant differences (H(2) = 1.726, *p* = 0.422, H(2) = 2.557, *p* = 0.278, H(2) = 2.040, *p* = 0.361, respectively); however, all post-treatment groups had significant differences (H(2) = 8.044, *p* = 0.018, H(2) = 7.441, *p* = 0.024, H(2) = 6.003, *p* = 0.050, respectively). Post hoc Mann–Whitney U tests using the Bonferroni-adjusted alpha level of 0.25 (0.05/2) were used to compare pairs of groups. For the first question, I feel that my advisor has appreciated me, there is no significant difference between the post-survey responses when comparing nature (Md = 4.00) and companion animal treatment (Md = 4.00) (U = 564.500, z = −1.955, *p* = 0.051, respectively). There was a significant difference in the post-survey results comparing companion animal treatment to control (Md = 3.50) (U = 534.000, z = −2.296, *p* = 0.022, respectively). For question two, I feel connected to my advisor, there were no statistical differences when comparing nature (Md = 3.00) and companion animal treatment (Md = 4.00) (U = 552.500, z = −2.043, *p* = 0.041, respectively; Bonferroni-adjusted alpha level of 0.25). There was a significant difference between the post-survey when comparing companion animal treatment and control (Md = 3.00) (U = 502.500, z = −2.592, *p* = 0.010).

For the last question, I believe my advisor cares about me, there were no statistical differences when comparing nature (Md = 3.50) and companion animal treatment (Md = 4.00) (U = 546.500, z = −2.160, *p* = 0.031, respectively; Bonferroni-adjusted alpha level of 0.25). There was a significant difference between the post-survey when comparing the companion animal treatment and control, (Md = 3.00) (U = 498.000, z = −2.693, *p* = 0.007).

#### Perceptions of Advisor and University

The hypothesis, companion animal stimuli prior to an advising session will increase students’ positive perspectives of their advisor in comparison to the control, was supported. However, there was no significant difference between the companion animal and nature treatments. As such, we are not able to fully support H_3_.

A question about comfort within the virtual advising setting was included in the modified UBQ but was not found to be specifically related to the advisor or university. It is important to note that a Kruskal–Wallis test was employed and that there was a significant difference in the pre-survey but no significant difference in the post-survey (H(3) = 8.792, *p* = 0.032; H(3) = 4.829, *p* = 0.185). The pre-survey wild animals (M = 3.34, Md = 4.00, SD = 0.966), companion animal (M = 3.39, Md = 4.00, SD = 0.916) and control (M = 3.53, Md = 4.00, SD = 0.725) had higher mean scores than nature (M = 3.08, Md = 3.00, SD = 0.969). Though not significantly different, in the post-survey all treatment mean scores increased: nature (M = 3.58, Md = 4.00, SD = 0.642), wild animal (M = 3.32, Md = 4.00, SD = 0.855), companion animal (M = 3.66, Md = 4.00, SD = 0.745) and the control did not (M = 3.45, Md = 4.00, SD = 0.921).

Animal stimuli prior to an advising session will not increase students’ positive perspectives of the university after advising sessions in comparison to the control and nature. In the case of the UBQ as an entire scale, it can be inferred that the hypothesis would be rejected given the information above. However, reviewing individual Likert scale items the hypothesis becomes harder to distinguish as seen in Table 4.

My university provides opportunities to engage in meaningful activities.I believe there are supportive resources available to me on campus.My university environment provides me an opportunity to grow.I believe I have enough academic support to get me through college.I am satisfied with my academic opportunities at my university.The university I attended values individual differences.

Given statement two and statement four had post-survey significance, a post hoc Mann–Whitney U was conducted using four different combinations of the post-survey treatment groups. As such, the Bonferroni-adjusted alpha level of 0.0125 (0.05/4) was used to compare pairs of groups. In statement two, the companion animal treatment (Md = 4.00) was compared to the nature treatment (Md = 3.00) (U = 548.000, z = −2.069, *p* = 0.039) the post test was not considered significantly different due to Bonferroni-adjusted alpha levels. Companion animal treatment was then compared to the control (Md = 3.00) (U = 506.000, z = −2.521, *p* = 0.012) and were considered significantly different in post-survey responses. After which, wild animal treatment was compared to the nature treatment and control. The wild animal treatment (Md = 4.00) was significantly different in post-survey responses to the nature and control treatment (U = 503.500, z = −2.625, *p* = 0.009, U = 468.500, z = −3.000, *p* = 0.003, respectively).

Statement four had similar results regarding the companion animal treatment (Md = 4.00) and the nature treatment (Md = 3.00) with no significant difference post-survey responses due to Bonferroni-adjustment alpha levels (U = 541.000, z = −2.136, *p* = 0.033). When the companion animal treatment was compared to the control (Md = 3.00) the results were not significant (U = 521.500, z = −2.340, *p* = 0.019). Similarly, wild animals (Md = 4.00) compared with nature had no significant difference in post-survey responses (U = 541.500, z = −2.130, *p* = 0.033). Following this pattern, there was no significant difference between the wild animal treatment and control (U = 523.500, z = −2.317, *p* = 0.021).

To provide a recap, we separated UBQ Likert scale questions to address university specific questions in order to provide more insight into our H_4._ The Kruskal–Wallis results indicated that only two of the six questions had significant differences between treatment groups. Post hoc Mann–Whitney U’s were conducted which indicate only one of the two questions had significant differences after the Bonferroni alpha level adjustment. As such, the only difference was found in the statement “I believe there are supportive resources available to me on campus” and this difference was significant for both animal type treatments and the control but only one animal treatment (wild animals) compared to nature. While supportive resources on campus refer to several different resources, it is likely that students may have associated this question with the advisor who is a valuable resource on campus; however, this is only an assumption. Given the context, these data allow us to partially support the hypothesis: Animal stimuli prior to an advising session will not increase students’ positive perspectives of the university after advising sessions in comparison to the control and nature.

### 3.5. Exploratory Analysis

After watching the randomized videos, each participant was asked four questions. Respondents were able to answer positive, negative or indifferent on the following question:How does this video make you feel?

Respondents were able to answer Yes or No to the following three questions:2.Did you like the content in this video?3.Would you be interested in watching similar videos before each advising session?4.Does this video increase your interest in the upcoming advising session?

The data were explored in a cross-tabulation analysis to provide information on the students’ perceptions of videos. While this is not data across pre-post measures, it expresses our populations’ interest in and influence of the content of the video they watched. For example, most students felt positive after watching a companion animal video compared to other videos. As such, it expresses that specific content may have made them feel more positive which is relevant to the H_1_ hypothesis. A χ-squared test was completed for all questions to determine if there were any significant differences across treatments. Regarding question 1, the χ2 test of independence was performed to examine the relationship between students’ feelings and the treatment video. The relationship between these variables was significant, χ2 (6, *N* = 154) = 43.509, *p* < 0.001. Table 5 shows the difference in students’ perception of the way the video made them feel.

Regarding questions 2–4, a χ^2^ test of independence was performed to examine the relationship between students’ feelings of like for the content, interest in watching similar videos, and if the video increased interest in the advising session with the treatment video. The relationship between each of these variables was significant (χ^2^ (3, *n* = 152) = 61.290, *p* < 0.001, χ^2^ (3, *n* = 152) = 17.629, *p* < 0.001, and χ^2^ (3, *n* = 152) = 13.170, *p* = 0.004, respectively).

As shown in Table 6, all students who watched a wild animal treatment liked the content of the video, a large majority liked companion animal and nature treatments and less than half of this population liked the content in the control video. Measures related to interest in similar videos before an advising session and increased interest in the upcoming advising session varied significantly compared to question 1 and 2. Almost half of the population was interested in watching similar videos before each advising session, more positive responses came from companion animal and wild animal treatments. In the case of increasing interest in the upcoming advising session, companion animal videos were the only videos to have more positive responses than negative responses which relays back to material in H_3_ and H_4,_ suggesting that companion animals play a role in priming social interaction. The data that support the findings of this study are available in Appendix A.

## 4. Discussion

According to the National Center for Education Statistics (NCES), 19.9 million students were enrolled at U.S. colleges [76]. The majority of full-time students in universities are between 18 and 24 years of age [77], which falls in the highest prevalence range for mental illness and disorders [78,79]. Almost 90 percent of students between the ages of 18 to 23 reported their education as a significant source of stress [80]. Despite this, according to the Association of University and College Counseling Center Directors, only 10.7% percent of students were served by their university counseling centers [81]. This may be because universities do not require students to meet with counselors or utilize stress-reduction resources. For individuals that do seek care, their needs may not be sufficiently met by their practitioner. According to a recent American Psychological Association survey, four in ten practitioners reported feeling burnt out, and three in ten felt they did not meet the demands for treatment [82]. However, many colleges require students to meet with their academic advisors [73]. Hence, the goal of this study was to explore a non-traditional and non-monetized opportunity within academia to implement programs aimed at addressing the need for mental health intervention and care provisioning in a non-clinical setting. Our findings indicated visual animal stimuli helped increase positive affect and that the companion animal treatment acted as social support prior to an advising session when compared to the control.

### 4.1. Positive Affect, Positive Emotions and Video Stimuli

Research has suggested that affect, which is a psychological state associated with emotions and mood, plays a critical role in linking psychological stress to disease. Long-term low positive affect has been associated with an increased risk of illness and mortality [83,84]. On the other hand, higher positive affect is linked to better health practices (i.e., improved sleep quality, more exercise, lower levels of stress hormones) and increases in health-relevant hormones (i.e., oxytocin) [85].

Our hypotheses stated that nature, wild animal and companion animal stimuli would increase positive affect in comparison to the control. The PANAS results indicate that participants who watched the companion animal treatment had a higher increase in positive affect scores than all other stimuli, including the control. This suggests that companion animals in comparison to wild animals and nature, are the most emotionally salient treatment type to increase positive affect. The findings indicate that companion animals were successful in increasing positive affect prior to an advising setting which is consistent with previous studies supporting the notion that interacting with animals increases positive affect [15,43,44,45].

In our second analysis, the mDES was employed to understand positive emotions. Emotions are commonly considered a subset of the broader class of affective phenomena including physical sensations, attitudes, moods, and even affective traits. Compared to the control, both wild animal and companion animal treatments increased students’ positive emotion, but nature did not in comparison to the control. Several explanations for this finding are put forth. The finding that all animal stimuli increased positive emotions suggests that animals may transmit positive emotions to viewers through emotional contagion, given that animals can express recognizable emotions [86]. This phenomenon may be relative to humans’ tendency to anthropomorphize animals (i.e., attributing human-like qualities and emotions to them). The evolutionary processes that led humans to be empathetic towards animals, a defining characteristic of our species [25,87], likely involved the development of a theory of mind (i.e., the ability to attribute mental states to oneself and others, including animals). This ability would have provided advantages for our ancestors in areas such as political alliances for mating [25] and emotionally nurturant caregiving for healthy developing children [39].

Further, the concept of baby schema (i.e., set of physical features, such as large eyes and round faces, associated with infant-like characteristics) often evoke caregiving behavior in humans and has been linked to positive emotions experienced in response to animals [42,88]. In contrast to animal stimuli, nature does not exhibit the same clear indicators of emotion or baby schema. Nature tends to evoke positive emotions through its beauty and grandeur, which are not necessarily tied to identifiable affective cues in the same way that animals can express emotions through facial expressions, and body language. Therefore, nature may not have been as effective in eliciting positive emotions in this study compared to the animal stimuli.

Positive emotions serve several important functions. Fredrickson’s broaden-and-build theory of positive emotions states that certain discrete positive emotions share the ability to broaden people’s momentary thought–action repertoire and build their personal resources (i.e., physical, intellectual, social, psychological) [58]. The theory suggests that unlike negative emotions, positive emotions are not intended to carry out adaptive benefits in situations that threaten survival but instead they carry indirect and long-term adaptive benefits by building enduring personal resources which can be drawn on when managing future threats.

What is important about this concept is that personal resources accrued during positive emotions are conceptualized as durable [89]. By broadening individuals’ momentary thought–action repertoires, it allows for openness to social, physical, intellectual and artistic behaviors [90,91]. An emotion such as interest creates the urge to explore, take in new information and experiences and expand oneself in the process [92,93,94,95]. These emotions outlast the transient emotional states that led to their acquisition allowing the student to increase their personal resources after the intervention and in subsequent moments when the students are meeting with their advisor. Further, this research implies students leave the meeting with positive emotions which then impact subjective well-being, coping, optimism and success, meaning that both companion animal and wild animal treatment have been used successfully in an intervention to improve health through a required setting in academia (academic advising).

### 4.2. Negative Affect, Negative Emotions and Video Stimuli

This study hypothesized that, compared to the control, all treatments would significantly reduce negative affect. However, the results showed that negative affect did not differ significantly between treatment types and the control group, unlike positive affect. Table 3 indicates that all treatments and the control group were effective in reducing negative affect, with mean decreases observed across all stimuli. While the decrease in negative affect on the PANAS or mDES cannot be attributed to the treatments, it is possible that the advisor played a crucial role in reducing negative affect across all groups. Advisors provide support, encouragement, and a safe space for students to express their feelings, which could have contributed to decreased negative affect across these groups.

### 4.3. Perception of Advisor and University

Positive affect has been shown in the literature to have a significant impact on social interaction, as individuals who experience positive affect are more likely to socialize regularly and maintain close social ties, leading to approach behavior [85]. Although participants who viewed companion animal videos reported a more positive perception of their academic advisor than the control group, the companion animal treatment did not outperform the nature treatment when increasing students’ positive perspective of their advisor due to Bonferroni correction, contrary to the hypothesis.

Research has suggested that both nature and companion animals can create calming environments [12,13,96], which could help students feel more at ease and willing to connect prior to an advising session. Further investigation is necessary to explore this relationship. Nonetheless, the findings of this study indicate that both were similarly effective in enhancing students’ perception of their advisors, which is a critical component of their sense of belonging at the university.

Additionally, student perception of the Webex setting was not significantly different between treatments. Treatments do transform the students’ perspective of their advisor, but they do not impact the students’ view of their university or the Webex setting. Notably, mean scores increased post advising sessions for all treatments except the control, although these increases were not statistically significant.

Further, treatments do transform the students’ perspective of their advisor, but they do not impact the students’ view of their university. It is likely that the advisor creates a positive relationship between the student and the university. The literature suggests academic advising plays a critical role in student satisfaction, engagement and retention at their university [74,75]. The advisor models a positive aspect of the university and therefore, can influence a student’s perception of their institution. Students that are more satisfied and engaged in their college experience are more likely to have a favorable perception of their institution. As seen in Table 4, all of the scores related to perception of the university increased regardless of treatment. Only one question “I believe there are supportive resources available to me on campus” had a significance between treatment types; however, it may be reasonable to assume that students are thinking of their advisor immediately after the advising session as a supportive resource available to them on campus. The data from this study would not support the idea that treatment types impacted students’ perception of the university.

### 4.4. Students’ Perception of the Treatment Stimuli

A cross tab analysis was used to analyze students’ perception of the treatment stimuli to help to interpret hypothesis-testing results of this study. The results showed that students felt most positive when watching the companion animals video, which is consistent with previous research suggesting that companion animals can foster emotional connections and empathy in humans. Recent studies conducted during the COVID-19 pandemic have suggested that individuals have turned to companion animals for increased social support and to alleviate mental and emotional stress, particularly those without children [97]. Although little research exists on the persistence of this human–pet bond post-pandemic, it is possible that this connection has endured and reflects in the effectiveness of the companion animal video stimuli in this study. Given that many undergraduate students likely do not have children and may be experiencing increased stress, the finding that the companion animal video was the most effective treatment in increasing positive affect and also in increasing positive perceptions immediately after receiving the treatment suggests that virtual animal interventions could be a valuable tool for improving mental health and well-being among university students.

In terms of respondents’ feeling towards the stimuli, the companion animal treatment was rated most positively, seconded by wild animal treatment, however, more respondents reported liking the wild animals treatment, followed by companion animals. Most individuals who watched the companion animal treatment were more interested in watching similar videos before an advising session, followed by the wild animal treatment. Lastly, the companion animal treatment was most effective in increasing students’ interest in the upcoming advising appointment. This information provides insight on students’ preferences for treatment content as well as what is emotionally relevant.

### 4.5. Implications and Future Directions

This study was not without limitations. Information relating to the study’s purpose was provided to both the advisor participant and students for their participation in the study. It was necessary to share the study’s purpose with the advising unit for seeking approval to conduct the study within this center and also to ensure data quality. Students were provided with this information for ethical compliance. Moreover, subjective responses may be susceptible to social desirability bias [98]. Additionally, because the survey consisted of 50 questions, in both the pre- and post-measures, students may have suffered from survey fatigue as well as performance while answering the questions. Furthermore, due to the nature of the design there was no way to ensure that participants had focused on their respective video during the pre-survey.

This study has implications for interventions that expand beyond AAI. The study’s findings suggest that virtual animal interventions can be applied to other settings involving client-serving industry. This study supports the potential of using a brief animal video before virtual advising sessions to enhance the virtual undergraduate advising experience in a university setting. Notably, the university where the study was conducted is a designated minority-serving institution, including Asian-American, Native-American, Pacific Islander-Serving Institution (AANAPISI) and Hispanic Serving Institution (HSI). This allowed the study to include a diverse sample population which is often challenging to recruit in HAI research [99]. The study highlights the importance of inclusivity in research and interventions targeting mental health. Moreover, the diversity of the sample population can increase the generalizability of the study’s findings beyond the university and shed light on how different populations may respond to the intervention, which could make the study’s framework and strategies more applicable in other settings and to a broader population.

Future work can build upon these initial results by testing causal connections in other advising centers. Additionally, future research should expand beyond advising and the university to explore this type of intervention prior to various service provider interactions (i.e., hospitality, healthcare offices, etc.) and may also build on individual items differences. For example, awe may be driving nature treatments and joy may be driving companion animal treatments to impact students’ perception of their advisor.

## 5. Conclusions

This is the first study to investigate the use of a required and accessible setting in academia (academic advising) as a means of care provisioning to target various aspects of mental health and well-being through randomly selected visual stimuli: nature, wild animal, companion animal and control. The study explores two main areas: the spaces that animals occupy as social medicine (positive social lubricants) for humans and the difference in affective responses to visual stimuli of wild animals, companion animals, nature, and control stimuli.

This study provided data on the use of one-minute companion animal and wild animal videos to promote positive affect among college students during their advising appointment. The findings indicate that exposure to companion animals in these videos was linked to the increase in students’ positive affect. Further, companion animal videos improved student’s perceptions of their advisors which can help strengthen the interaction between a student and their advisor. These computer-mediated interventions fill an important service gap to improve well-being outcomes when individuals do not have access to physical assistance, and may ultimately have a broader impact outside of advising and the university.

## Figures and Tables

**Table 1 animals-13-01522-t001:** Descriptive Participant Socio-demographic Characteristics.

Variable	*n*	Mean (SD) or %
Age	152	20.07 (3.853)
Gender		
Female	82	53.9%
Male	64	42.1%
Genderqueer	3	2.0%
Prefer not to respond	3	2.0%
Ethnicity/Race		
American Indian or Alaskan Native	2	1.3%
Asian or Pacific Islander	29	19.1%
Black or African American	20	13.2%
Hispanic	53	34.9%
Middle Eastern	2	1.3%
Multi-racial	13	8.6%
White or Caucasian	29	19.1%
Prefers not to respond	4	2.6%
Class Level		
Incoming Freshman	8	5.3%
Freshman	80	52.6%
Sophomore	51	33.6%
Junior	10	6.6%
Senior	1	0.7%
Non-degree seeking	2	1.3%
Exploring Major or Decided on Major		
Exploring	65	42.8%
Decided	87	57.2%
Of those Decided:		
Science Pathways	29	19.1%
Engineering Pathways	16	10.5%
Business Pathways	23	15.1%
None of these	19	12.5%

**Table 2 animals-13-01522-t002:** Correlation Matrix.

Variable	Pre PANAS-P	Post PANAS-P	Pre PANAS-N	Post PANAS-N	Pre mDES-P	Post mDES-P	Pre mDES-N	Post mDES-N	Pre-UBQ
Pre- PANAS-P	1	--	--	--	--	--	--	--	--
Post- PANAS-P	0.654 **	--	--	--	--	--	--	--	--
Pre- PANAS-N	0.211 **	−0.011	--	--	--	--	--	--	--
Post- PANAS-N	0.127	−0.051	0.704 **	--	--	--	--	--	--
Pre-mDES-P	0.812 **	0.604 **	0.168 *	0.145	--	--	--	--	--
Post- mDES-P	0.606 **	0.889 **	0.063	−0.027	0.668 **	--	--	--	--
Pre-mDES-N	0.006	−0.061	0.700 **	0.618 **	0.126	−0.018	--	--	--
Post-mDES-N	0.097	−0.063	0.630 **	0.862 **	0.122	−0.058	0.608 **	--	--
Pre-UBQ	0.349 **	0.220 **	0.065	0.116	0.360 **	0.245 **	0.016	0.139	--
Post-UBQ	0.299 **	0.431 **	0.000	0.019	0.257 **	0.426 **	−0.095	−0.014	0.681 **

* Correlation is significant at the 0.05 level (2-tailed); ** Correlation is significant at a 0.01 level (2-tailed).

**Table 3 animals-13-01522-t003:** Mean, Standard Deviations, and Range PANAS and mDES Variables.

Variable	Treatment Type Mean ± SD (Range)			
	Nature	Wild Animal	Companion Animal	Control
Pre-PANAS-P	3.3184 ± 0.96586	3.2132 ± 0.84026	3.30 ± 0.95776	2.9816 ± 0.89378
	(1.10–4.80)	(1.30–4.60)	(1.00–4.80)	(1.00–4.20)
Post-PANAS-P	3.5053 ± 0.97453	3.5921 ± 0.78581	3.6974 ± 0.83972	2.9763 ± 1.07689
	(1.80–5.00)	(1.40–4.90)	(1.70–5.00)	(1.00–4.70)
Pre-mDES-P	3.2211 ± 1.1121	3.3684 ± 1.0460	3.5553 ± 0.90425	2.8895 ± 1.0170
	(1.00–4.90)	(1.00–4.90)	(1.20–4.90)	(1.00–4.40)
Post-mDES-P	3.6132 ± 0.88963	3.8868 ± 0.76765	3.9289 ± 0.77666	3.1211 ± 1.1395
	(1.90–5.00)	(1.90–5.00)	(1.80–5.00)	(1.00–4.90)
Pre-PANAS-N	1.9921 ± 0.71410	1.8632 ± 0.74595	1.5508 ± 0.60616	1.7895 ± 0.75473
	(1.10–3.80)	(1.00–3.90)	(1.00–3.10)	(1.00–4.20)
Post-PANAS-N	1.5079 ± 0.61748	1.5132 ± 0.57289	1.4974 ± 0.59525	1.5684 ± 0.68066
	(1.00–3.30)	(1.00–3.10)	(1.00–3.40)	(1.00–3.40)
Pre-mDES-N	1.9632 ± 0.68079	1.6921 ± 0.51485	1.8395 ± 0.59482	1.8132 ± 0.78436
	(1.00–3.90)	(1.00–2.60)	(1.00–3.10)	(1.00–4.10)
Post-mDES-N	1.4211 ± 0.53684	1.4474 ± 0.54313	1.4316 ± 0.57097	1.5132 ± 0.59054
	(1.00–2.90)	(1.00–3.10)	(1.00–3.10)	(1.00–3.20)

**Table 4 animals-13-01522-t004:** Kruskal–Wallis Test Results for pre- and post-survey responses comparing treatment differences via UBQ Likert Scale questions.

Statement	Treatment	Survey	Mean ± SD	H	*p*
My university provides opportunities to engage in meaningful activities.	Nature	Pre	2.95 ± 0.928	2.799	0.424
		Post	3.34 ± 0.669	5.209	0.157
	Wild Animal	Pre	3.13 ± 0.935		
		Post	3.37 ± 0.852		
	Companion Animal	Pre	3.24 ± 0.852		
		Post	3.53 ± 0.762		
	Control	Post	3.53 ± 0.762		
		Pre	3.11 ± 0.764		
I believe there are supportive resources available to me on campus	Nature	Pre	3.05 ± 0.985	4.266	0.234
		Post	3.37 ± 0.663	13.741	0.003
	Wild Animal	Pre	3.34 ± 0.966		
		Post	3.61 ± 0.885		
	Companion Animal	Pre	3.37 ± 0.883		
		Post	3.58 ± 0.758		
	Control	Pre	3.24 ± 0.820		
		Post	3.13 ± 0.963		
My university environment provides me an opportunity to grow.	Nature	Pre	2.92 ± 0.969	4.952	0.175
		Post	3.37 ± 0.633	5.476	0.140
	Wild Animal	Pre	3.08 ± 0.882		
		Post	3.45 ± 0.860		
	Companion Animal	Pre	3.32 ± 0.873		
		Post	3.50 ± 0.797		
	Control	Pre	3.13 ± 0.777		
		Post	3.18 ± 0.896		
I believe I have enough academic support to get me through college.	Nature	Pre	2.95 ± 0.899	3.118	0.374
		Post	3.34 ± 0.627	10.112	0.018
	Wild Animal	Pre	3.21 ± 0.935		
		Post	3.53 ± 0.862		
	Companion Animal	Pre	3.18 ± 0.865		
		Post	3.55 ± 0.765		
	Control	Pre	3.08 ± 0.882		
		Post	3.18 ± 0.896		
I am satisfied with the academic opportunities at my university	Nature	Pre	2.84 ± 0.916	4.676	0.197
		Post	3.29 ± 0.732	6.124	0.106
	Wild Animal	Pre	3.16 ± 0.973		
		Post	3.45 ± 0.891		
	Companion Animal	Pre	3.16 ± 0.855		
		Post	3.50 ± 0.797		
	Control	Pre	3.00 ± 0.805		
		Post	3.18 ± 0.896		
The university I attended values individual differences	Nature	Pre	2.97 ± 0.915	2.063	0.559
		Post	3.32 ± 0.702	3.893	0.273
	Wild Animal	Pre	3.08 ± 0.912		
		Post	3.34 ± 0.847		
	Companion Animal	Pre	3.24 ± 0.886		
		Post	3.53 ± 0.797		
	Control	Pre	3.24 ± 0.820		
		Post	3.26 ± 0.891		

Note: *n =* 38 for all of the above variables.

**Table 5 animals-13-01522-t005:** Cross Tabulation Analysis on Question 1.

Question	Treatment	Positive	Negative	Indifferent
		*n* (%)	*n* (%)	*n* (%)
How did this video make you feel?	Nature	25 (26.0%)	2 (40.0%)	11 (21.6%)
	Wild Animal	28 (29.2%)	0	10 (19.6%)
	Companion Animal	34 (35.4%)	0	4 (7.8%)
	Control	9 (9.4%)	3 (60.0%)	26 (51.0%)
Total		96 (100.0%)	5 (100.0%)	51(100.0%)

**Table 6 animals-13-01522-t006:** Cross Tabulation Analysis on Questions Two–Four.

Questions	Treatment	Yes	No
		*n* (%)	*n* (%)
Did you like the content in this video?	Nature	34 (89.5%)	4 (10.5%)
	Wild Animal	38(100.0%)	0
	Companion Animal	37 (97.4%)	1 (2.6%)
	Control	15 (39.5%)	23 (60.5%)
Total		124 (100.0%)	28 (100.0%)
Would you be interested in watching similar videos before each advising session?	Nature	19 (50.0%)	19(50.0%)
	Wild Animal	21 (55.3%)	17 (44.7%)
	Companion Animal	24 (63.2%)	14 (36.8%)
	Control	7 (18.4%)	31 (81.6%)
Total		71 (100.0%)	81 (100.0%)
Does this video increase your interest in the upcoming advising session?	Nature	17 (44.7%)	21 (55.3%)
	Wild Animal	18 (47.4%)	20 (52.6%)
	Companion Animal	22 (57.9%)	16 (42.1%)
	Control	7 (18.4%)	31 (81.6%)
Total		96 (100.0%)	5 (100.0%)

## Data Availability

Data are contained within the Appendix A.

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
