# Peer review of "Examining the Impact of Virtual Animal Stimuli on College Students’ Affect and Perception of their Academic Advising Experience"

_animals, 2023, doi:10.3390/ani13091522_

Round 1

Reviewer 1 Report

Review of “Examining the Impact of Virtual Animal Stimuli on College Students’ Affect and Perception of their Academic Advising Experience”

The authors conducted a study where 152 college students were given various videos including videos of companion animals to watch and measured the impact of the type of video on various measures of mental well-being, as well as student perception of their school and advisor.

Introduction

Line 47, I think it would be fairer to say that there is a correlation between stress and a lower GPA and lower academic performance. Which is the cause and which is the effect does not follow from the cited source, and thus it remains to be seen whether attenuating stress will result in better academic performance or will merely result in students feeling better about their lack of academic success.

Line 117, be more specific in how indirect exposure improves participants’ physiological health.

Line 129, this reads like there may be something missing between “animals [40, 41],” and “this”.

Line 137, “veterinary students” is sufficient and more concise – though it might be worth pointing out that veterinary students are probably not representative of the general population when it comes to a positive attitude towards animals.

Line 165, is “animal” here meant to include both wild and domesticated animals? Please clarify.

M&M

Please also provide IRB approval and informed consent information as part of this section.

Line 186, it seems somewhat problematic that the advisors were informed about the purposes of the study, which may have biased their own behavior during the appointments. Same comment for Line 198, psychological studies are generally conducted in such a way as to keep participants in the dark regarding the study purpose so that they are not biased in the relevant outcome domains.

Line 256 and following, provide references for these surveys (compare comment for Measures section, the progression of the paper is off in this regard).

Line 275 and following include source URL’s for the videos here or as an appendix.

Line 285 and following, this doesn’t belong in the methods section, please move to the introduction or the discussion.

Measures section, it seems like the surveys used should be described and referenced before their administration is discussed, please move this section in front of line 251 and following, or mention that they are explained further down in the M&M section.

Statistics section, place and emphasize the explanation of your correction for multiple comparisons more prominently.

Results

At first glance these results appear female and minority skewed, is that correct? Is this representative of your general student population, and if not, what could have caused it?

As above, I’m not seeing any indication of correcting for multiple comparisons, please comment.

Line 490, what was the composition of that principal component? How does it relate to real-world outcomes?

Lines 556 and following, do we know whether answers to these subjective statements have any influence on measurable criteria of academic success?

Line 619, I think it would be more meaningful to give percentages of viewers of each video rather than percentage of viewers who reacted in a certain way across videos.

Discussion

As in the introduction, there needs to be at least some emphasis on how these emotional outcomes affect academic performance, as well as an explicit discussion of the question which is the cause and which is the effect.

Line 694 et al. provide a reference for the baby schema concept.

Reviewer 2 Report

This was a very well writen article and provided an interesting way to support students in the new digital age.  I only saw minor wording isues where the language was a bit awkward (e.g. p3, lines 125-126; p5, line 220; p6, lines 260-261; p6, line 263), but nothing of significance.  Great job!

Reviewer 3 Report

One of the worries I have about this paper is that there is no mention of either the bak ground or the interests of the students. It may be that those who found companion animals a help had them themselves or were very familiar with them and those that did not did not. Also those who are studying biological subjects might well have preferred "nature". IT might be that those particularly interested in the internet preferred the control too! May be they would all have enjoyed having a puzzle on the internet, rather than just videos of virtual reality?

I am slightly worried also about the consequences of this type of approach, are we in the future going to rely on virtual companion animals or nature or other humans to service our emotional needs in the future? Why was the advisor not also virtual? 

Some rewriting and shortening would be good. Emotionality is a private thing so trying to measure it is of course difficult, and again I am not convinced that the quoted references actually understand this. Passed experiences will greatly effect what ever emotion is being experience, whether virtual or real!
